# Preventing and treating PTSD-like memory by trauma contextualization

Alice Shaam Al Abed [1,2,3], Eva-Gunnel Ducourneau[1,2], Chloé Bouarab[1,2,4], Azza Sellami [1,2], Aline Marighetto[1,2,5] & Aline Desmedt [1,2,5 ✉]

Post-traumatic stress disorder (PTSD) is characterized by emotional hypermnesia on which preclinical studies focus so far. While this hypermnesia relates to salient traumatic cues, partial amnesia for the traumatic context can also be observed. Here, we show in mice that contextual amnesia is causally involved in PTSD-like memory formation, and that treating the amnesia by re-exposure to all trauma-related cues cures PTSD-like hypermnesia. These findings open a therapeutic perspective based on trauma contextualization and the underlying hippocampal mechanisms.

[1] Neurocentre Magendie, Physiopathologie de la plasticité neuronale, U1215, INSERM, 33000 Bordeaux, France. [2] Université de Bordeaux, 33000 Bordeaux, France. [3]Present address: Eccles Institute of Neuroscience, John Curtin School of Medical Research, The Australian National University, Canberra, ACT, Australia. [4]Present address: The George Washington University, Ross Hall—Room 7272, 300 Eye (I) Street, NW, Washington, DC 20037, USA. [5]These authors contributed equally: Aline Marighetto, Aline Desmedt. ✉email: aline.desmedt@inserm.fr

Post-traumatic stress disorder (PTSD) is a devastating psychiatric disorder that develops after a traumatic event experienced as a threat of injury or death, with a lifetime prevalence of about 8%[1]. PTSD patients experience recurrent and intrusive recollection of traumatic memories characterized by intense fear responses in ordinary, safe situations (i.e. emotional hypermnesia), while having difficulties retrieving exhaustive narrative (i.e. declarative) memories of their trauma, as certain aspects of the context are missing[1–6]. This contextual amnesia, believed to result from hippocampal hypofunction induced by intense stress[7–13], is most frequently partial, and even very discreet sometimes. Nevertheless, certain clinicians suggest that this amnesia might play a role in the development and persistence of intrusive recollections of traumatic memories[4–6]. Namely, the memory deficit for peritraumatic contextual cues would impair the capability of the subject to restrict fear to the traumatic place and cues. Decontextualized, traumatic memories would escape voluntary control as they would be automatically reactivated, potentially in whatever context, by the sole presence of salient cues more or less related to the trauma[14]. Yet, the potential role of amnesia in PTSD[8,9,11,12] has been left unexplored, as current research essentially focuses on the most obvious memory symptom: emotional hypermnesia.

We here hypothesized that contextual amnesia and the hippocampal dysfunction believed to cause it constitute a critical factor in the etiology of PTSD-related hypermnesia and its persistence. We validated this hypothesis using the first animal (mouse) model that recapitulates the two memory components, i.e. hypermnesia and amnesia of PTSD[15]. In this model, traumatic stress is mimicked by the combination of contextual fear conditioning with post-conditioning intra-peritoneal injection of corticosterone (CORT), the main stress hormone in rodents[13,16] (Fig. 1, top panel). Contextual fear conditioning is achieved through a tone–shock unpairing paradigm, meaning that the tone, even though salient, does not predict the shock delivery. Hence, a normal, adapted memory of the conditioning is characterized by a strong fear response to the conditioning context but not to the irrelevant tone. In contrast, CORT-injected mice display an abnormal fear response to the tone, hence mimicking the pathological emotional hypermnesia. In parallel, when re-exposed to the conditioning context alone (i.e., Context test), CORT-injected mice display a reduced fear response to the conditioning context that reveals a contextual amnesia.

Using this model, we tested the hypothesis that contextual amnesia is causally involved in PTSD-like memory by manipulating contextual memory formation through optogenetic inhibition/activation of the hippocampus. Namely, we tested whether hippocampal inhibition during conditioning, that should impair context memorization, promotes the formation of PTSD-like memory in Vehicle (Veh)-injected mice, whereas hippocampal activation, expected to promote context memorization, may prevent PTSD-like memory in CORT-injected mice.

## Results

Mice received a bilateral injection of AAV5-expressing ArchT (inhibitory channel) or ChR2 (excitatory channel) into the dorsal CA1 (dCA1), and then received chronic bilateral optic fiber implants. dCA1 was targeted because (i) it was previously shown to be hypoactivated in mice developing PTSD-like memory compared to those expressing normal fear memory[15] and (ii) it is classically described as a key hippocampal subfield for both contextual and declarative memory[17,18].

First, in control animals that were not submitted to optogenetic stimulation (light off) during conditioning (Day 1), memory tests (Day 2) confirmed our previous findings[15]: CORT-injected mice displayed PTSD-like alterations of fear memory with an abnormal fear response to the tone and decreased fear to the context (Fig. 1a, top). Second, the inhibition of dCA1 neurons during conditioning was sufficient to induce PTSD-like memory in Veh-injected mice. Indeed, ArchT Veh-injected mice conditioned with light on displayed not only a lower fear response to the context compared to those trained with light off, but also an abnormal fear response to the tone, i.e. a memory profile similar to that of CORT-injected mice (Fig. 1a, top). In contrast with dCA1 inhibition, similar manipulation of dCA2 or dCA3 failed to produce PTSD-like fear memory (Fig. 1a, bottom). dCA3 inhibition only tended to induce an abnormal fear response to the tone, without any effect on the fear response to the context. Third, in contrast to inhibition, optogenetic activation of dCA1 neurons in the trauma-inducing condition (i.e., CORT-injected mice) prevents the formation of PTSD-like memory (Fig. 1b). Indeed, ChR2 CORT-injected mice trained with light on during conditioning, in contrast to those trained with light off, did not display any (abnormal) fear response to the tone and expressed a normally high fear response to the context. Finally, the effects of optogenetic inhibition or activation were all specifically due to dCA1 manipulation during the stressful episode since the same manipulations were ineffective when performed 5 min before fear conditioning (Fig. 1c). Altogether, our first experiment demonstrates that amnesia for the traumatic context, due to hippocampal hypoactivity during the trauma, is causally involved in the development of PTSD-like memory, and that this pathological process can be prevented by promoting contextual memorization through hippocampal activation. This observation echoes with clinical observations suggesting that mental narrative of the event during the trauma, an active attitude likely associated to hippocampal activation and better contextualization, prevents the development of PTSD[19].

In a clinical perspective though, it is crucial to know whether contextualization of the trauma could cure PTSD-related memory once it has been developed. We reasoned that retrieving traumatic memories while being in the traumatic context may promote a "re-contextualization" of the trauma, and thereby restore normal fear memory. To test this hypothesis, after conditioning (+CORT or Veh injection, Day 1) and the first memory tests (Day 2), mice were re-exposed to the conditioning context on Day 3 in the presence of the tone, which is the salient cue to which the trauma has been (abnormally) associated. PTSD-like alterations of fear memory (i.e. abnormal fear to the tone + reduced fear to the context) observed in CORT-injected mice on Days 2 (Fig. 1) and 3 (Fig. 2a, left) disappeared after simultaneous re-exposure to all (tone + context) trauma-related cues. CORT-injected mice indeed behaved like controls on Day 4 (Fig. 2a, right). In contrast, partial re-exposure to trauma-related cues, to the tone alone (Fig. 2b), or to the conditioning context alone (Fig. 2c), or to both the tone and the context but spaced out of 2 h (Fig. 2d) failed to modify the traumatic memory, which remained PTSD-like on Day 4. The fact that mice display relatively high level of freezing in the neutral context may indicate that animals generalize their fear from the conditioning to the neutral context, and that consequently a partial trauma re-contextualization may be engaged. However, despite this high level of freezing which is classically observed in all mice in this first memory test after fear conditioning, the fact that "PTSD-like" mice re-exposed to the tone in the neutral context still display a PTSD-like memory the day after indicates that re-exposure to this neutral/familiar context does not induce the re-contextualization of traumatic memory. Noticeably in patients, trauma re-exposures that are generally incomplete are also of mixed efficacy[20]. Here in mice, reactivation of traumatic memories normalizes these memories insofar as this reactivation occurs in the traumatic context but not in a neutral context (i.e. incomplete reactivation), indicating that "re-contextualization" of the trauma is the curative factor. The fact that

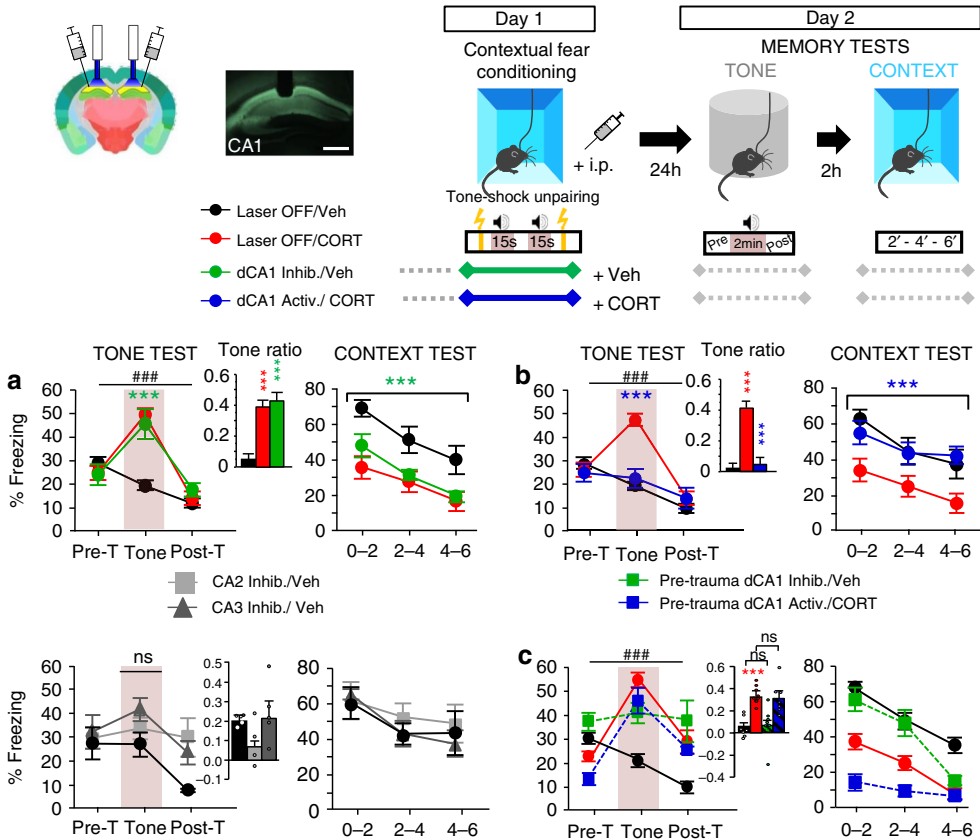

**Fig. 1 dCA1 inhibition during stress produces PTSD-like memory whereas dCA1 activation in traumatic conditions prevents PTSD-like memory.** Top panel, scheme illustrating the optogenetic approach (Image credit: Allen Institute), contextual fear conditioning (Day 1) and memory tests (Day 2): re-exposure to the tone alone in a familiar chamber, and then to the conditioning context 2 h later. Scale bar: 500 μm (**a**) top, compared to normal contextual fear memory attested by no fear response to the tone (no increase of freezing to the tone, left panel) and high fear response to the context (right panel) in (Laser OFF, $n = 15$) Vehicle-injected mice, PTSD-like memory in (Laser OFF, $n = 13$) CORT-injected mice is attested by a fear response specific to the tone (repeated measures (RM) of freezing during the tone test: $F_{2,24} = 35.908$; $P < 0.0001$; tone ratio vs 0; $P < 0.0001$) associated with decreased fear to the context. A similar difference between the groups is observed when NaCl is replaced by HBC as vehicle [data not shown]. dCA1 inhibition during conditioning produces PTSD-like memory in Veh-injected mice ($n = 13$; RM × laser condition: $F_{4,76} = 20.184$; $P < 0.0001$). Although all groups display relatively high pre-tone freezing levels in the familiar chamber, these levels are lower than those expressed in the conditioning chamber (left vs. right: all $F > 6.29$, all $P < 0.028$). **a**, bottom, dCA2 or dCA3 inhibition does not produce PTSD-like memory ($n = 4–5$; laser condition: $F_{2,11} = 1.755$; $P = 0.2181$). **b** Compared to control (Laser OFF; $n = 11$) CORT-injected (PTSD-like) mice, CORT-injected mice submitted to dCA1 activation ($n = 11$) during conditioning display a normalized (contextual) fear memory (RM × laser condition: $F_{2,40} = 15.684$; $P < 0.0001$). **c** Pre-conditioning (5 min before) dCA1 inhibition or activation does not change the nature of the fear memory formed: normal (contextual) or PTSD-like fear memory in Veh- and CORT-injected mice, respectively ($n = 7–8$). Data are presented as mean ± SEM. This experiment was repeated independently once with similar results. See table in the "Methods" section for detailed sample sizes. Statistical significance was assessed by RM (three blocks) two-sided ANOVA with post hoc test when appropriate. ***$P < 0.005$. ###: block × condition interaction ($P < 0.005$). Source data are provided as a Source Data file.

"PTSD-like" mice discriminate these two contexts, as attested by a higher level of freezing in the conditioning than in the neutral context, indicates that a specific processing of the traumatic context makes the re-contextualization process possible when traumatic memory is reactivated in this context.

We next demonstrated that this curative effect is hippocampus-dependent. As shown in Fig. 3, optogenetic inhibition of dCA1 during re-exposure to the tone in the conditioning context completely abolishes the curative effect of re-exposure. ArchT CORT-injected mice with light on during re-exposure still displayed PTSD-like memory on Day 4, in contrast to their controls with light off.

Finally, PTSD is a long-lasting stress-related disorder, and in a therapeutic perspective, it is essential to show that the expression of "re-contextualized" traumatic memory remains durably normal. Therefore, 1 month after conditioning we tested the persistence of PTSD-like and "re-contextualized" fear memories. As shown in Fig. 4, the "re-contextualized", and thus normalized, traumatic memory remained undistinguishable from normal memory (Veh-injected controls, Fig. 4a), and PTSD-like memory observed after dCA1 inhibition during complete re-exposure or partial re-exposure to traumatic cues persisted as such (Fig. 4b–d).

## Discussion

This study provides the first evidence that contextual amnesia is causally involved in the development and persistence of PTSD-like emotional hypermnesia, thereby demonstrating what some clinical studies had suggested so far[4–6,14]. Thus, inducing contextual amnesia by dCA1 inhibition during a stressful event is sufficient to induce the hypermnesia component of PTSD-like memory, whereas in traumatic conditions, promoting memorization of the context through dCA1 activation enables the formation of normal memory, and thus prevents the development of PTSD-like memory. How can we explain this hippocampus-dependent switch between normal and PTSD-like fear memory?

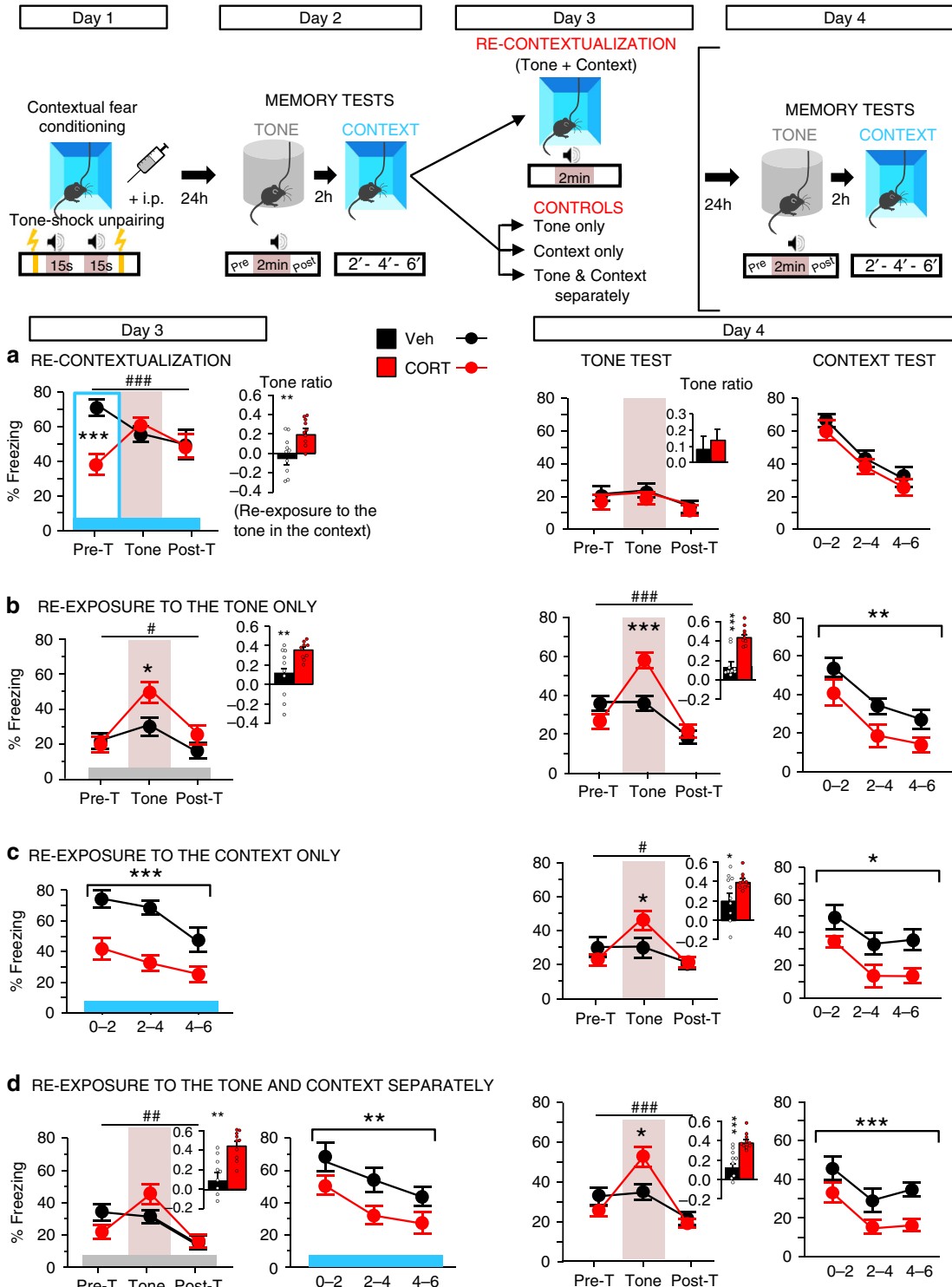

In line with contemporary conditioning theories, we previously showed that contextual and elemental (discrete tone) conditioning are each based on an imbalance between two (hippocampus- or amygdala-based) memory systems and that manipulations of the dorsal hippocampus can affect in an opposite way tone and contextual fear memory[21,22]. Here, knowing that stress, or glucocorticoids exposure, can impair hippocampal plasticity[13,23,24], we suggest that (i) CORT injection promoted a cognitive imbalance in favor of an elemental (amygdala-based) tone fear conditioning and that (ii) hippocampal activation might have thwarted some of the CORT-induced deleterious effects on the

hippocampus, thereby restoring a cognitive imbalance in favor of context processing.

In a therapeutic perspective, the present study also reveals that once PTSD-like memory is developed, enabling the hippocampus-dependent "re-contextualization" of the trauma through re-exposure to all trauma-related cues durably restores normal fear memory expression. This normalization of fear memory may be mediated by the reconsolidation process according to which reactivated memory traces become transiently labile and thus sensitive to manipulations[25–27]. However, in contrast to normal fear memory for which frequent reactivation

**Fig. 2 Switching from PTSD-like memory to normal fear memory by trauma re-contextualization.** Top panel, scheme illustration of the behavioral procedure: fear conditioning (Day 1), memory tests (Day 2), trauma re-contextualization by complete re-exposure to trauma-related cues (i.e. to the tone in the conditioning context vs. partial re-exposure to the tone alone, context alone, or tone and context spaced out of 2 h; Day 3), memory tests (Day 4). **a** Curative effect of trauma re-contextualization with a switch from PTSD-like memory on Day 3 to normal contextual memory on Day 4: complete re-exposure to trauma-related cues in CORT-injected mice abolishes the abnormal fear response to the tone and normalizes the conditioned fear to the context (Veh vs CORT: ns). **b–d** Partial re-exposure to trauma-related cues (tone alone, context alone or tone and context spaced out of 2 h) on Day 3 does not modify the PTSD-like memory profile in CORT-injected mice, which still display an abnormal fear response to the tone together with a low conditioned fear to the context on Day 4 (minimum significance in the tone test: repeated measures × Veh/CORT: $F_{2,36} = 4.755$; $P = 0.0147$; in the context test: Veh/CORT: $F_{1,18} = 4.570$; $P = 0.0465$). CORT-injected mice discriminate the two contexts used, as attested by their higher freezing levels in the conditioning context than in the familiar chamber (first 2 min, 2a vs. 2b; $F_{1,17} = 7.166$; $P = 0.0159$). The blue and gray bars symbolize the conditioning context and the familiar/safe chamber, respectively; Pre-T: Pre-Tone; Post-T: Post-Tone. Data are presented as mean ± SEM. This experiment was repeated independently once with similar results, see Table 1 in the "Animals" section of the methods for detailed sample size ($n = 9$–10). Statistical significance was assessed by repeated measures (three blocks) two-sided ANOVA. $*P < 0.05$; $**P < 0.01$; $***P < 0.005$. #: block × condition interaction ($P < 0.05$); ##$P < 0.01$; ###$P < 0.005$. Source data are provided as a Source Data file.

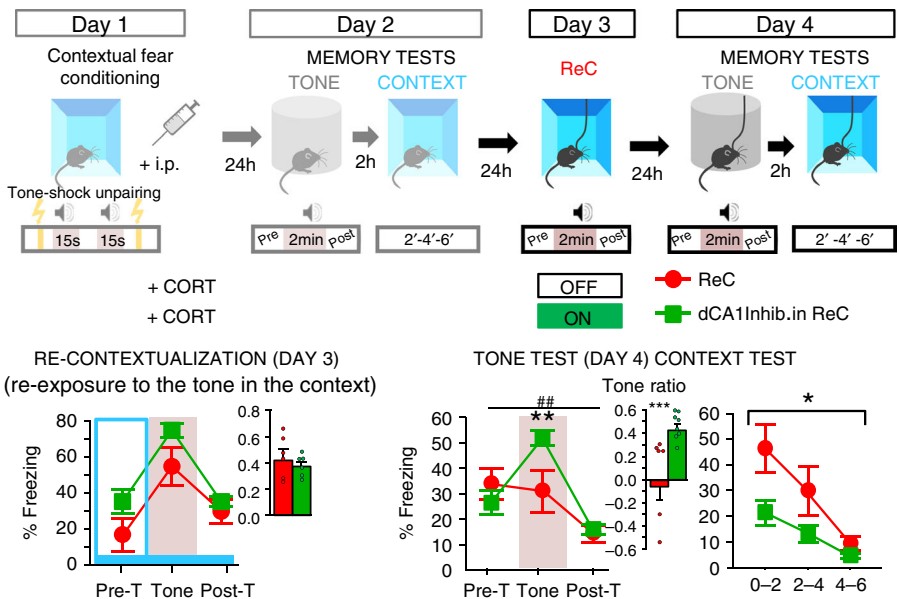

**Fig. 3 Trauma re-contextualization is hippocampus-dependent.** Top panel, scheme illustration of the behavioral procedure: fear conditioning (Day 1), memory tests (Day 2), trauma re-contextualization (ReC) by complete re-exposure to trauma-related cues (i.e. to the tone in the conditioning context; Day 3) in control CORT-injected and CORT-injected mice exposed to dCA1 inhibition during re-exposure, memory tests (Day 4). Bottom, dCA1 inhibition during complete re-exposure to trauma-related cues on Day 3 blocks the curative effects of such re-exposure: compared to their control (PTSD-like) re-exposed with Laser OFF (ReC; $n = 6$), PTSD-like re-exposed animals with Laser ON ($n = 7$) still display an abnormal fear response to the tone (repeated measures × laser condition: $F_{2,24} = 6.452$; $P = 0.0057$) together with a low conditioned fear to the context (laser condition: $F_{1,12} = 6.922$; $P = 0.0219$) on Day 4. The blue bar symbolizes the conditioning context; Pre-T: Pre-Tone; Post-T: Post-Tone. Data are presented as mean ± SEM. This experiment was repeated independently once with similar results, see table in the "Animals" section of the "Methods" for detailed sample size. Statistical significance was assessed by repeated measures (three 2 min-blocks) two-sided ANOVA. $*P < 0.05$; $**P < 0.01$. ##: block × condition interaction ($P < 0.01$). Source data are provided as a Source Data file.

in safe contexts classically leads to its reduction, frequent reactivation of PTSD-related memory through flashbacks in safe contexts seems to contribute to its persistence. We suggest that reactivation of traumatic memory in PTSD would be incomplete. As suggested by clinicians, the disconnection of PTSD-related memory from context-based representations would prevent re-evaluation of the traumatic event, thus strengthening the original memory trace[6,19]. In contrast, the re-appraisal of the traumatic event in the light of its contextual representation would promote its integration into the declarative memory system, leading thereby to recovery from PTSD-related memory[6,19]. Present demonstration that recovery from PTSD-like hypermnesia depends on recovery from contextual amnesia calls for promoting therapeutic approaches of PTSD centered on trauma contextualization and its underlying hippocampal mechanisms.

## Methods

**Animals.** For all the experiments, 3-month-old naive male mice (C57Bl/6j, Charles River) were individually housed in standard Makrolon cages in a temperature and humidity controlled room under a 12-h light/dark cycle (lights on at 07:00) and had ad libitum access to food and water. All experiments took place during the light phase. Every effort was made in order to minimize the number of animals (cf. Table 1) used and their suffering. All experimental procedures were conducted in accordance with the European Directive for the care and use of laboratory animals (2010-63-EU) and the animals care guidelines issued by the animal experimental committee of Bordeaux University (CCEA50, agreement number A33-063-099; authorization No. 01377). The group sizes were as follows:

Experiment 1 [optogenetic inhibition/activation of dCA1 (vs. dCA2 and dCA3) during fear conditioning (vs. before conditioning)].
Experiment 2 (trauma "re-contextualization").
Experiment 3 (optogenetic inhibition of dCA1 during trauma "re-contextualization").
Experiment 4 (assessment of the persistence of PTSD-like and "re-contextualized" fear memory on Day 30): these mice are the same as in Experiments 2 and 3.

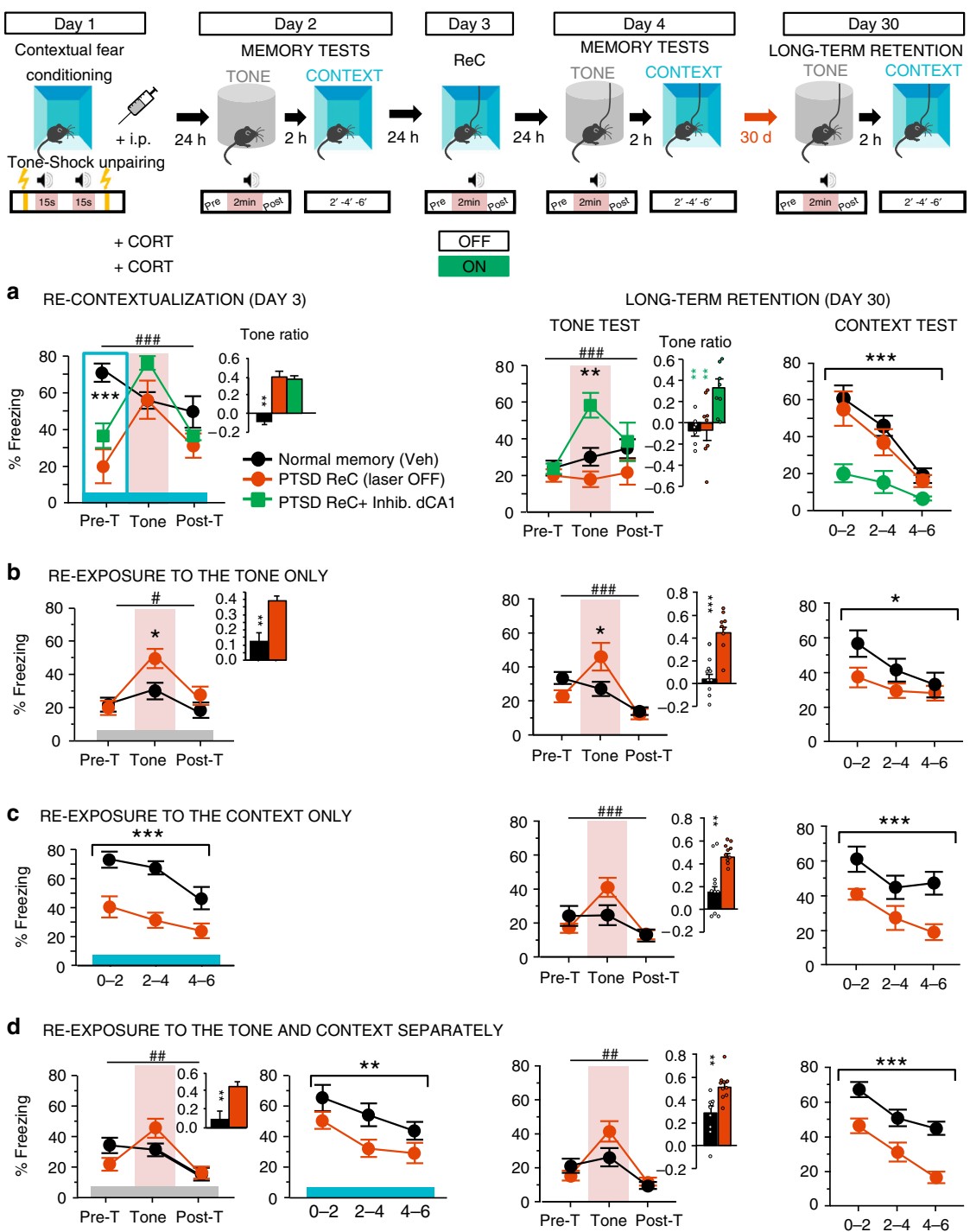

**Fear conditioning procedure**. The day before fear conditioning, all mice were individually placed for 4 min into a round chamber (20 cm diameter) with an opaque PVC floor, in a brightness of 100 lux. The box was cleaned with 1% acetic acid before each trial. This pre-exposure allowed the mice to acclimate and become familiar with the chamber later used for the tone re-exposure test. Acquisition of fear conditioning (Day 1) was performed in a different context, a squared conditioning chamber (24 × 24 cm), in a brightness of 100 lux, given access to the different visual-spatial cues in the experimental room. The floor of the chamber consisted of 19 stainless-steel rods (3 mm diameter), spaced 1 cm apart and connected to a shock generator. The box was cleaned with 70% ethanol before each trial. All animals were trained with a tone–shock unpairing procedure, meaning that the tone was non-predictive. This training procedure, routinely used in our laboratory, promotes the processing of contextual cues in the foreground[15,21,22,28]. Briefly, each animal was placed in the conditioning chamber for 4 min during which it received two tone cues (65 dB, 1 kHz, 15 s) and two foot-shocks (squared signal: 0.4 mA, 50 Hz, 1 s) according to a pseudo-randomly distribution. Specifically, 100 s after being placed in the chamber, animals

received a shock, then, after a 20 s interval, a tone; finally, after a 30 s delay, the same tone and the same shock spaced by a 30 s interval were presented. After 20 s, animals were returned to their home cage. In this tone–shock unpairing procedure, as the tone is never followed by shock delivery, animals identify the conditioning context (set of static background contextual cues that constitutes the environment in which the conditioning takes place), and not the tone, as the right predictor of the shock (Fig. 1, top, Day 1).

Twenty-four hours after the acquisition of fear conditioning, mice were submitted to two memory retention tests (Day 2). During these two memory tests, animals were continuously recorded for off-line second-by-second scoring of freezing by an observer blind of experimental groups. Freezing behavior of animals, defined as a lack of all movement except for respiratory-related movements, was used as an index of conditioned fear response. Mice were first submitted to the tone re-exposure test in the safe familiar chamber during which three successive recording sessions of the behavioral responses were performed: one before (first 2 min), one during (next 2 min), and one after (2 last min) tone presentation (Fig. 1, top, Day 2). Conditioned

**Fig. 4 PTSD-like and normalized fear memory are persistent.** Top panel, scheme illustration of the behavioral procedure: fear conditioning (Day 1), memory tests (Day 2), trauma re-contextualization (ReC) by complete re-exposure to trauma-related cues (vs. partial re-exposure to the tone alone, context alone, or tone and context spaced out of 2 h; Day 3) in control Veh-injected ($n = 9$), control CORT-injected ($n = 8$) and CORT-injected mice exposed to dCA1 inhibition during re-exposure ($n = 7$), memory tests (Day 4 and Day 30). **a** The normalized contextual fear memory (attested by the absence of fear response to the tone together with high conditioned fear to the context) induced by trauma re-contextualization (PTSD ReC) and the remained PTSD-like memory (i.e. attested by abnormal fear response to the tone and low conditioned fear to the context) subsequent to dCA1 inhibition during re-contextualization persist for at least 30 days after trauma (Tone test: repeated measures × laser condition: $F_{4,42} = 4.717$; $P = 0.0031$; Context test: laser condition: $F_{2,21} = 5.489$; $P = 0.0121$; PTSD ReC vs CA1 Inhib: $P = 0.253$). **b–d** The remained PTSD-like memory subsequent to partial re-exposure to trauma-related cues persists whatever the re-exposure condition (to the tone alone, context alone, or tone and context spaced out of 2 h; minimum significance in the tone test: repeated measures × Veh/CORT: $F_{2,34} = 7.935$; $P = 0.0015$; in the context test: Veh/CORT: $F_{1,17} = 4.607$; $P = 0.0466$, repeated measures × Veh/CORT: $F_{2,34} = 4.266$; $P = 0.0022$). The blue and gray bars symbolize the conditioning context and the familiar/safe chamber, respectively; Pre-T Pre-Tone, Post-T Post-Tone. Data are presented as mean ± SEM. This experiment was repeated independently once with similar results, See table in the "Animals" section of the "Methods" for detailed sample size. Statistical significance was assessed by repeated measures (three 2 min-blocks) two-sided ANOVA with post hoc test when appropriate. *$P < 0.05$; **$P < 0.01$; ***$P < 0.005$. #: block × condition interaction ($P < 0.05$); ##$P < 0.01$; ###$P < 0.005$. Source data are provided as a Source Data file.

**Table 1 Detailed number of mice in each experimental group.**

|  | Groups | Final number | Starting number |
|---|---|---|---|
| EXPERIMENT 1 | OFF/Veh | 8 (Fig. 1b)–15 (Fig. 1a) | 8 (Fig. 1b)–15 (Fig. 1a) |
|  | OFF/CORT | 11 (Fig. 1b)–13 (Fig. 1a) | 12 (Fig. 1b)–15 (Fig. 1a) |
|  | dCA1 activ./CORT | 11 | 14 |
|  | dCA1 inhib./Veh | 13 | 15 |
|  | Pre-trauma dCA1 activ./CORT | 7 | 8 |
|  | Pre-trauma dCA1 inhib./Veh | 7 | 8 |
| EXPERIMENT 2 | Fig. 2a Veh—ReC | 10 | 10 |
|  | Fig. 2a Cort—ReC | 10 | 10 |
|  | Fig. 2b Veh—Tone only | 10 | 10 |
|  | Fig. 2b Cort—Tone only | 9 | 9 |
|  | Fig. 2c Veh—Context only | 9 | 9 |
|  | Fig. 2c Cort—Context only | 9 | 9 |
|  | Fig. 2d Veh—Tone + Context | 9 | 9 |
|  | Fig. 2d Cort—Tone + Context | 10 | 10 |
| EXPERIMENT 3 | PTSD ReC— Laser OFF | 7 | 7 |
|  | PTSD ReC—Inhib CA1 | 8 | 8 |
| EXPERIMENT 4 | PTSD ReC—Laser OFF | 7 | 7 |
|  | PTSD ReC—Inhib CA1 | 7 | 7 |
|  | Other groups | Same number as Exp 2 | Same number as Exp 2 |

response to the tone is expressed by the percentage of freezing during the tone presentation compared to the levels of freezing expressed before and after tone presentation (repeated measures on three blocks of freezing).The strength and specificity of this conditioned fear is attested by a ratio that considers the percentage of freezing increase to the tone with respect to a baseline freezing level (i.e., pre- and post-tone periods mean). Indeed, a strong and specific conditioned fear to the discrete tone CS implies a lower level of freezing when the shock is not expected (i.e. 2 min before and 2 min after the tone presentation) compared to the freezing level expressed during the tone presentation (high ratio value). The tone ratio is calculated as follows: [% freezing during tone presentation − (% pre-tone period freezing + % post-tone period freezing)/2]/[% freezing during tone presentation + (% pre-tone period freezing + % post-tone period freezing)/2]. Two hours later, mice were submitted to the context re-exposure test: they were placed for 6 min in the conditioning chamber. Freezing to the context was calculated as the percentage of the total time spent freezing during the successive three blocks of 2-min periods of the test. While the first block is the critical block attesting difference between animals that are conditioned to the conditioning context and those that are not or less, the following two blocks are presented in order to assess a gradual extinction of the fear responses in the absence of shock. Normal contextual fear memory is attested by a high conditioned fear to the conditioning context (right predictor of the shock) together with an absence of conditioned fear to the non-predictive tone. In contrast, a maladaptive PTSD-like fear memory is attested by an opposite pattern of results indicating the erroneous selection of the tone instead of the context as predictor of the shock: an abnormal fear response to the tone associated with a decreased conditioned fear to the context[15].

**Systemic injection of corticosterone.** Corticosterone (2-hydroxypropyl-β-cyclodextrin complex; 2.5 mg/kg in a volume of 0.1 ml/10 g bodyweight) or vehicle (NaCl 0.9%) was administrated intraperitoneally (i.p.) immediately after the acquisition of fear conditioning. The complex of corticosterone with cyclodextrin allows dissolving this steroid in aqueous solutions. After the injection, animals were returned to their home cage. The dose of corticosterone was selected on the basis of previous results indicating that such dose (i) is in the range of concentrations induced by stress in the plasma[29,30] and (ii) effectively induces PTSD-like memory in mice when combined with fear conditioning using a relatively high footshock intensity (squared signal, 0.4 mA)[15].

**Trauma re-contextualization (Experiment 2).** A decontextualized memory of the trauma, associated with the systematic avoidance of conscious recollection of the trauma, would contribute to the intrusive re-experiencing of the trauma in safe situations (i.e. flashbacks) characterizing PTSD[4,5,11]. Accordingly, treatment of PTSD would imply a switch from a decontextualized intrusive traumatic memory to a normal, contextualized, fear memory. We thus reasoned that re-exposure to the most emotionally laden traumatic cue (i.e. the tone in our study) in the traumatic context could not only induce trauma reminder but also promote the re-contextualization of the trauma and thereby cure PTSD-like memory. Two days after fear conditioning (Day 3), animals were thus re-exposed to the tone cue in the conditioning context (Fig. 2, top, Day 3). Three successive recording sessions of the behavioral responses were performed: one before (first 2 min), one during (next 2 min), and one after (2 last min) tone presentation. Therefore, the first 2 min (pre-tone) allowed us to assess the level of conditioned fear to the conditioning context alone, while the conditioned response to the tone was assessed during the next 2 min, both by the percentage of freezing during the tone presentation and by the tone ratio described above. Three control conditions have been designed: either re-exposure to the tone alone (in the familiar context), or to the conditioning context alone, or both to the tone and to the context but spaced out of 2 h. The effects of these different re-exposure conditions to trauma-related stimuli on fear memory were assessed on day 4 when animals were successively re-exposed to the tone cue in the familiar context, then to the conditioning context 2 h later (same tests as in Day 2). While an abnormal fear response to the tone together with a low conditioned fear to the context would confirm PTSD-like memory, an absence of

**Table 2 Detailed description of the optogenetic manipulations for each experimental group in Experiment 1 (Day 1).**

| Optogenetic manipulation | | | Group |
|---|---|---|---|
| **Day 1** | | | |
| **Before** | **Conditioning** | **i.p. injection** | |
| **Experiment 1** | | | |
| No Laser | No Laser/ArChT | Veh (NaCl) | OFF/NaCl |
| No Laser | No Laser/ChR2 | CORT | OFF/CORT |
| No Laser | ON/ChR2 (activ.) | CORT | dCA1 Activ./CORT |
| No Laser | ON/ArChT (inhib.) | Veh (NaCl) | dCA1 Inhib./NaCl |
| ON/ChR2 (activ.) | No Laser | CORT | Pre-trauma dCA1 Inhib./NaCl |
| ON/ArChT (inhib.) | No Laser | Veh (NaCl) | Pre-trauma dCA1 Activ./CORT |

conditioned fear to the tone together with a high conditioned fear to the context would attest for a normalization of traumatic memory.

**Optogenetic manipulations of dCA1 or dCA2/dCA3 activity**. Mice underwent a two-step surgery 4 weeks before the beginning of behavior. First, mice were bilaterally injected with AAV5 expressing ArChT to inhibit glutamatergic neurons (AAV-CaMKIIα-ArchT-GFP, UNC Vector Core) or ChR2 to activate glutamatergic neurons (AAV-CaMKIIα-ChR2(H134R)-EYFP, UNC Vector Core), using glass pipettes (tip diameter 25–35 μm) connected to a picospritzer (Parker Hannifin Corporation) into the dCA1 at two injection sites to minimize diffusion to extra dCA1 areas (0.2 μl each site; P1: AP −1.8 mm; L ± 1.3 mm; DV −1.4 mm/P2: AP −2.5 mm; L ± 2 mm; DV −1.4 mm, according to a classical stereotaxic procedure). Maximum and minimum area of virus injection from the anterior dCA1 (Bregma −1.34) to the posterior dCA1 (Bregma −2.7) are represented in Supplementary Fig. 1. Second, mice were implanted with bilateral optic fiber implants (diameter: 200 μm; numerical aperture: 0.39; flat tip; Thorlabs) directed to the dCA1 (AP: −1.8, L: ±1.3, DV: −1.4). Implants were fixed to the skull with Super-Bond dental cement (Sun Medical, Shiga, Japan). Correct placements of fibers were visually checked on hippocampal slices to reject all mice with fiber located outside the medial part of anterior dorsal CA1. A picture representative of the virus injection[31] and of the fiber position in dCA1 is presented in Fig. 1. In experiments targeting dCA2 or dCA3 to assess the area selectivity of the effects observed with dCA1 manipulations, the procedure was identical except coordinates of virus injection (0.2 μl each side, AP: −2.0, L: ±2.5, DV: −2) and fiber implantation (AP: −1.8, L: ±2.1, DV: 1.7 for dCA2; AP: −1.85, L: ±2.45, DV: −1.9 for dCA3).

For every optogenetic manipulations, the light (approximately 6 mW per implanted fiber) was bilaterally conducted from the laser (473 or 526 nm, CNI) to the mice via two optic fiber patch cords (diameter 200 μm, Thorlabs), connected to a rotary joint (intensity splitter rotary joint, Doric Lenses) that allowed mice to freely move in the behavioral apparatus. For inactivation (ArChT), the light was continuously delivered at 526 nm and for activation (ChR2), the light was delivered at 473 nm, 5 Hz (5195). Previous observations indicate that light stimulation per se does not impact the observed freezing behavior. Indeed, previous optogenetic experiments achieved in our laboratory clearly showed that the freezing behavior of control GFP mice (Laser ON) was not impacted by light stimulation of the dCA1 neurons[17].

**Experiment 1**. The aim was to assess whether contextual amnesia, and the hippocampal hypofunction believed to cause it, may be responsible for the development of PTSD-like memory. Indeed, we previously showed that the imbalance between tone and contextual conditioning is critically dependent on the hippocampal–amygdalar circuit. More specifically, we showed that altering the hippocampal function (e.g. cholinergic transmission) can modulate the amygdalar activation and promote tone fear conditioning to the detriment of contextual conditioning[21]. By another way, we also showed that CORT-induced PTSD-like memory is associated with hippocampal hypoactivation[15], while several other studies showed that stress or glucocorticoids exposure can alter the dorsal hippocampus, resulting in neuronal loss, decrease in dendritic branching, alterations in synaptic terminal structure, inhibition of neuronal regeneration, impairment of long-term potentiation and in a reduction in the production of new neurons in this brain region[13,23,24]. We thus reasoned that under relatively high stressful situation, the optogenetic inhibition of dCA1, which was shown to reduce c-Fos expression in this hippocampal sub-region[32] as it is observed after CORT injection[15], should promote a cognitive imbalance in favor of an elemental (tone) fear learning in a situation in which the context is yet the main predictor of the threat (maladaptive fear memory).

Mice were submitted to the fear conditioning procedure described above and we either inhibited or activated the pyramidal cells of the hippocampus during the whole conditioning session. The optogenetic manipulation "during" fear conditioning was preferred because in PTSD patients, while an impaired narrative (i.e. declarative) memory of their trauma is observed after the traumatic event, a deficit in mental narrative of the event during the trauma appears as a risk factor for the development of PTSD[19]. Since CORT-inducing PTSD-like memory is

**Table 3 Detailed description of the optogenetic manipulations in Experiment 3 (Day 3).**

| Optogenetic manipulation | Group |
|---|---|
| **Day 3** | |
| **Re-contextualization** | |
| **Experiment 3** | |
| No Laser | ReC |
| ON/ArChT (inhib.) | dCA1 Inhib. in ReC |

administered immediately after fear conditioning[15] the optogenetic inhibition of dCA1 cells may also be efficient if performed "after" conditioning, during the consolidation phase of fear memory. However, the necessarily time-limited optogenetic inhibition of dCA1 cells can hardly recapitulate the deleterious (both functional and structural, cf. above) effects of post-training CORT injection on memory consolidation. Indeed, insofar as (i) memory consolidation and CORT effects can both extend over several hours, the precise time window during which the optogenetic manipulation should be performed during the (24 h) retention interval is difficult to predict and (ii) optogenetic manipulation cannot be performed during the entire 24 h retention interval, we thus chose not to manipulate the dCA1 (also) after fear conditioning. In the first experiment, we assessed the involvement of dCA1 compared to dCA2 and dCA3 (see optogenetic manipulations of dCA1 or dCA2/dCA3 activity in the Methods section). Because only the inhibition of dCA1 impacted the fear memory, the rest of the experiments was focused on the manipulation of dCA1. To make sure that the critical parameter was the manipulation of dCA1 "during" the fear conditioning, we added two control groups, which were submitted to the same protocol but had their dCA1 only activated or inhibited "before" the conditioning. On Day 1, all mice were submitted to the same protocol (Fig. 1, top): they were put in a chamber where they were attached to the laser. Depending on the experimental group (cf. Table 2), they were either subjected to activation or inhibition of dCA1, or had no laser on. This step lasted the same amount of time as the acquisition of fear conditioning. They were immediately put in the conditioning chamber and submitted to the fear conditioning session. The next day, fear memory was tested as described above.

**Experiment 3**. To assess the involvement of dCA1 in the re-contextualization process, we added a group of mice (cf. Table 3) that were submitted to the same procedure described in the trauma re-contextualization section, but had their dCA1 pyramidal cells optogenetically inhibited during the whole re-contextualization session (i.e. re-exposure to the tone in the conditioning chamber). The next day (Day 4), mice were again submitted to the two memory tests (Fig. 3).

**Experiment 4: Assessment of the persistence of "re-contextualized" and not "re-contextualized" PTSD-like fear memory**. In order to demonstrate that the curative effect of trauma re-contextualization on PTSD-like memory was long-lasting, animals that recovered from PTSD-like memory on Day 4 after having been submitted to complete re-exposure to trauma-related cues (i.e. tone in the conditioning context on Day 3) were again submitted to two memory tests (tone test and context test spaced out of 2 h, as in Day 2) 30 days after fear conditioning. Similarly, animals that still displayed PTSD-like fear memory on Day 4 after having been subjected to the inefficient partial re-exposure (i.e. to the tone only, context only, or tone and context spaced out of 2 h), or to the optogenetic inhibition of dCA1 cells during the re-contextualization session, were submitted to the same memory tests in order to test the persistence of their PTSD-like memory.

**Statistics**. Data are presented as the mean ± SEM error bar. Statistical analyses were performed using analysis of variance (ANOVAs) followed by Fisher's PLSD post hoc test when appropriate. Analyses were performed using StatView software. Statistical significance was considered at $P < 0.05$.

**Reporting summary**. Further information on research design is available in the Nature Research Reporting Summary linked to this article.

## Data availability

All data are available in the main text or the Supplementary Information. Source data are provided with this paper.

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

## Acknowledgements

We thank Cyril Herry (Neurocentre Magendie) for the development of optogenetic tools in our laboratory. We thank all of the personnel of the Animal Facility of the Neurocentre Magendie for mouse care. This work was supported by Centre National de la Recherche Scientifique, Institut National de la Santé et de la Recherche Médicale, Fondation pour la Recherche sur le Cerveau, Conseil Régional d'Aquitaine, Ministère de l'Enseignement supérieur et de la Recherche, University of Bordeaux and the Agence Nationale pour la Recherche Grant PTSDmemo (ANR-14-CE13-0026-01), as well as an EquipEx Grant OptoPath (ANR-10-EQX-008-1).

## Author contributions

A.M. and A.D. conceived, designed, and supervised the experiments. A.S.A.A. performed the experiments. A.S.A.A. and A.D. analyzed the data. A.S.A.A., C.B., A.S., E.-G.D., A.M., and A.D. discussed the results. A.S.A.A., A.M., and A.D. wrote the manuscript.

## Competing interests

The authors declare no competing interests.
