## [Peer Review File · Nature Communications]

Reviewers' Comments:

Reviewer #1:

Remarks to the Author:

The authors tested a hypothesis that hippocampal hypofunction-associated contextual fear amnesia contributes to PTSD-like 'decontextualized' (generalized) emotional hypermnesia. To test this, a series of experiments were conducted in mice subjected to unpaired tone and footshock stimuli followed by systemic corticosterone injection (to produce PTSD-like memory generalization to the tone), and then their contextual memory strengths were altered via and optogenetic inhibition/activation of the dorsal CA1 hippocampus. The main findings are that (i) optogenetic inhibition of CA1 during (but not prior to) fear conditioning produced generalized fear to the tone stimulus (akin to posttraining corticosterone injection), (ii) optogenetic stimulation of CA1 neutralized corticosterone-induced tone fear memory, and (iii) retrieving generalized tone fear memory in the fear conditioned context ('re-contextualization') restored normal fear memory. Based on these findings, the authors conclude that "dCA1 inhibition during a stressful event is sufficient to induce both contextual amnesia and hypermnesia components of PTSD-like memory" and that "promoting memorization of the context through dCA1 activation prevents the development of PTSD-like memory." In my view, the findings are novel, well presented and possibly translationally significant. However, I do have some comments and questions.

1. The pre-tone freezing levels during the tone test in a familiar chamber are comparably high in all groups (e.g., Fig. 1a,b; Fig. 3), which would indicate generalized fear between trained and familiar chambers (both are round with 20 cm diameter). If so, shouldn't there be some re-contextualization going on during the tone test?
2. The post-training cort injection producing PTSD-like effects require some explanation. For example, is the cort effect due to its action on dCA1 cells since the dCA1 inhibition during fear conditioning mimics post-training cort effect, whereas the dCA1 activation offsets post-training cort effect? If so, this should be tested by applying optogenetic inhibition, not only 5 minutes before and during fear conditioning (as shown in Fig 1b,c) but also right after fear condition.
3. Related to comment #2, there should be some mentioning of how cort affects CA1 neurons, and thereby the formation of tone fear memory, which is presumably either hippocampal-independent or ventral hippocampal-dependent.
4. (Fig 1a, Tone Test) For the Laser OFF/Veh group, the % freezing during the Tone period is lower than % freezing during the Pre-T period. If so, why is the Tone Ratio shown as being greater than 0? This seems to be the case for all Tone Ratio figures shown in Figure 1.
5. Are the n sizes for the four experiments before or after histological verification? Providing both before and after histology n size information would be helpful.
6. How was the freezing behavior quantified by a blind observer? Time sampling or duration?
7. Shouldn't the vehicle be β -cyclodextrin dissolved in .9% NaCl, rather than NaCl alone, to confirm that β -cyclodextrin is inert?

Reviewer #2:

Remarks to the Author:

Alice Shaam Al Abed et al investigated roles of hippocampus in generation of PTSD-like memory and switching this memory to normal fear memory by trauma-contextualization and re-contextualization using optogenetics. The authors showed that optogenetic inactivation of hippocampus during the exposure to the cue is sufficient to generate PTSD-like memory while activation of hippocampus with CORT injection blocks the formation of PTSD-like memory. Interestingly, reactivation of traumatic memory by re-exposure to the context with the cue switched the PTSD-like memory to normal fear memory and this normalization was hippocampus-dependent. Furthermore, the authors showed the persistence of PTSD-like and normalized memories. Importantly, the findings support the importance of exposure therapy for PTSD that may normalize PTSD-like memory through reactivation of this memory. However, the novelty of findings in this manuscript seems to be not so much because the authors have already shown that the generation of PTSD-like memory is hippocampus-dependent using microinfusion of CORT into hippocampus (ref15).

My other major concerns are as follows.

- 1) Optogenetic activation of hippocampal CA1 in the presence of CORT blocked generation of PTSD-like memory. This observation is interesting. However, it still remains unclear why this artificial hippocampal activation blocks generation of PTSD-like memory by CORT. The authors should show or suggest how this activation could block the effect of CORT.
- 2) In figure1, several control experiments are required such as mice expressing GFP with laser and mice expressing

ChR2/ArchT without laser.

3) Normalization of PTSD-like memory must be mediated by memory updating processes such as reconsolidation, extinction and/or re-learning. Although it is interesting to examine these mechanisms, at least, the authors should discuss about these possibilities.

Description of revisions and reply to the reviewers' concerns:

General:

We were pleased to learn that both reviewers considered the present findings of considerable potential interest, the first one indicating that these findings “*are novel, well presented and possibly translationally significant*”, the second one highlighting the fact that they “*support the importance of exposure therapy for PTSD that may normalize PTSD-like memory through reactivation of this memory*”.

Nevertheless, both reviewers propose ways to improve our paper, for which we are grateful. The requested changes have been included all over the manuscript (cf. text highlighted in yellow in the revised manuscript) as requested by the reviewers' recommendations.

Our point-by-point responses to the reviewers' comments (which are now *fully written below in italics and in quotation marks*) are detailed below:

Reviewer 1:

“(Remarks to the Author):

The authors tested a hypothesis that hippocampal hypofunction-associated contextual fear amnesia contributes to PTSD-like ‘decontextualized’ (generalized) emotional hypermnnesia. To test this, a series of experiments were conducted in mice subjected to unpaired tone and footshock stimuli followed by systemic corticosterone injection (to produce PTSD-like memory generalization to the tone), and then their contextual memory strengths were altered via and optogenetic inhibition/activation of the dorsal CA1 hippocampus. The main findings are that (i) optogenetic inhibition of CA1 during (but not prior to) fear conditioning produced generalized fear to the tone stimulus (akin to posttraining corticosterone injection), (ii) optogenetic stimulation of CA1 neutralized corticosterone-induced tone fear memory, and (iii) retrieving generalized tone fear memory in the fear conditioned context (‘re-contextualization’) restored normal fear memory. Based on these findings, the authors conclude that “dCA1 inhibition during a stressful event is sufficient to induce both contextual amnesia and hypermnnesia components of PTSD-like memory” and that “promoting memorization of the context through dCA1 activation prevents the development of PTSD-like memory.” In my view, the findings are novel, well presented and possibly translationally significant. However, I do have some comments and questions.”

1. Is there any re-contextualization during the tone test (in the familiar context)?

“The pre-tone freezing levels during the tone test in a familiar chamber are comparably high in all groups (e.g., Fig. 1a,b; Fig. 3), which would indicate generalized fear between trained and familiar chambers (both are round with 20 cm diameter). If so, shouldn't there be some re-contextualization going on during the tone test?”

This relatively high pre-tone freezing level (about 25-30%), classically observed in all groups, is likely a *basal level of freezing* in this test rather than a generalized fear between the trained and the familiar context. Indeed,

- First, this test being *the first one performed 24h after the acquisition of fear conditioning*, it is also the first time animals are carried back into the same experimental room used for fear conditioning the day before (even if the particular context of the test is a familiar one) and by the same experimenter: these conditions likely explain a relatively high basal level of freezing during the two first minutes of this first memory test.

-Second, this relatively high level of freezing constitutes a *starting level* either for the expression of a much higher freezing response to the tone which has been abnormally associated with the

footshock in CORT-injected mice and in NaCl-injected mice submitted to dCA1 inhibition, or for a decrease of the freezing response in control mice for which the tone does not predict the footshock.

-Finally, the fact that “PTSD-like” mice re-exposed to the tone in the familiar context still display a PTSD-like memory the day after (i.e. abnormally high fear response to the tone and impaired fear response to the context) (Fig. 2b) indicates that re-exposure to the tone in the familiar context does not induce the re-contextualization of traumatic memory in contrast to the re-exposure to the tone in the conditioning context.

All in all, these observations make very unlikely a generalization of the fear between the conditioning and the familiar context, and more definitively, the experiment 2 (cf. Fig. 2b) shows that trauma re-contextualization does not occur when animals are re-exposed to the familiar context.

This important conclusion has been emphasized in the revised version of the manuscript (cf. p. 4-5).

2. The post-training cort injection producing PTSD-like effects requires some explanation.

“The post-training cort injection producing PTSD-like effects require some explanation. For example, is the cort effect due to its action on dCA1 cells since the dCA1 inhibition during fear conditioning mimics post-training cort effect, whereas the dCA1 activation offsets post-training cort effect? If so, this should be tested by applying optogenetic inhibition, not only 5 minutes before and during fear conditioning (as shown in Fig 1b,c) but also right after fear condition.”

We thank the reviewer for raising this important concern.

Yes we do believe that the effect of post-training peripheral injection of CORT is due to its deleterious impact on the dorsal hippocampus-dependent formation of declarative/contextualized fear memory. The main support for this view is the fact that CORT injected directly into the dorsal hippocampus has the same (PTSD-like memory inducing) effect as peripheral injection of CORT as shown in our original paper (Kaouane et al., Science, 2012). In fact, when combined with fear conditioning using a US of sufficiently high intensity (as in the present study), post-training corticosterone injection mimics traumatic stress. This is further confirmed by the fact that the deleterious effect of CORT (intra-dHPC or systemic injections) can be more physiologically reproduced by a post-training restraint stress (Kaouane et al, 2012). In contrast, when combined with fear conditioning using a US of low intensity, post-training CORT mimics moderate stress and in that case, promotes/enhances dHPC-dependent contextualized fear memory (as we previously showed in Revest et al, Nat. Neurosci., 2005, and Kaouane et al, Science, 2012).

Since CORT is administered immediately after fear conditioning, it must affect the consolidation phase of dHPC-dependent memory. This implies that optogenetic inhibition of dCA1 cells here found to produce the same effects as post-training CORT injection when performed *during* (but not *before*) fear conditioning, should also be efficient if performed *after* conditioning, during the consolidation phase of fear memory. However, (i) the *precise time window* during which the optogenetic manipulation should be performed during the (24 hour) retention interval is difficult to predict since memory consolidation and CORT effects can both extend over several hours, and (ii) optogenetic manipulation cannot be performed during the entire 24h retention interval. That is why we initially chose not to manipulate the dCA1 (also) *after* fear conditioning. Besides, in humans, a deficit in mental narrative of the event *during* the trauma seems to be a risk factor for the development of PTSD (Brewin, 2011), which is why the optogenetic manipulation of the dCA1 *during* fear conditioning was initially chosen.

Nevertheless, as requested by the Reviewer, we conducted a new optogenetic experiment in which animals were submitted to a dCA1 inhibition (ArchT NaCl) of short duration *right after fear conditioning* (4 min similar to the manipulation performed during conditioning) combined with a systemic injection of NaCl, in order to test whether such inhibition could also mimic the traumatic effect of post-training CORT injection. Two GFP control groups (GFP NaCl & GFP CORT) were included. The results are presented in the following figure. Not only the dCA1 inhibition failed to reproduce the effects of CORT injection compared to NaCl injection (seen in animals that did not receive any optogenetic manipulation: see top right corner of each panel), but all 3 groups submitted to

post-conditioning, pseudo (GFP) or effective (ArchT) optogenetic inhibition, exhibited similarly low but significant conditioned fear response to the tone and to the context, indicative of some fear generalization in all groups. Altogether, these findings suggest that the 10 min period immediately following fear conditioning might be particularly sensitive to any form of additional stress, even though the deleterious effect of peripheral CORT injection on dHPC-dependent memory consolidation likely extends above this sensitive period.

Figure 1. Conditioned freezing to the tone and to the context in mice previously submitted to optogenetic inhibition of dCA1 right after fear conditioning. **a.** During re-exposure to the tone (in the familiar context), all groups display significant conditioned freezing to the discrete tone [global effect of the tone (ANOVA, MR, 3 blocks: $F(2,42) = 30.998$, $p < 0.0001$) that does not differ among groups (block x group: $F(4,42) = 0.549$, $p = 0.70$)]. In contrast, in controls (not subjected to optogenetic manipulation, top right corner), only CORT-injected mice display a significant tone fear conditioning (block x group: $F(2,28) = 37.364$, $p < 0.0001$) **b.** During re-exposure to the context, all groups display similarly low conditioned fear to the context (absence of group effect: $F(2,42) = 0.032$, $p = 0.968$). In contrast, CORT-injected control mice (top right corner) display lower levels of conditioned fear than NaCl-injected control mice ($F(1,28) = 11.318$, $p = 0.0046$)

We think these inconclusive results should not be presented in the final version of the manuscript as they would blur the main message of the paper, i.e. the fact that hippocampal inhibition *during* stress can cause PTSD-like memory, whereas hippocampal activation prevents its formation. Nevertheless, this figure may be presented as supplementary information if requested.

In addition, the technical reasons for which we chose not to manipulate the hippocampus after conditioning, together with the scientific reasons for which we expected significant effects of hippocampal manipulations performed *during* conditioning are now explicitly mentioned in the method section of the revised manuscript (cf. p. 15).

3. How corticosterone affects the formation of tone fear memory?

“Related to comment #2, there should be some mentioning of how cort affects CA1 neurons, and thereby the formation of tone fear memory, which is presumably either hippocampal-independent or ventral hippocampal-dependent.”

We show that corticosterone impairs the consolidation of declarative/contextualized fear memory known to rely on dorsal hippocampus, but also promotes the formation of elemental/cue conditioning known to rely on the ventral hippocampus and amygdala. These opposite effects of corticosterone on two different forms of fear memory are in line with current literature showing that the different memory systems are not independent but interact (compete or cooperate) in different manners depending on the situation, including stress level. Even though the mechanisms underlying the effects of corticosterone in the present study remain to be specified, we can speculate that the impact of corticosterone on tone fear memory is an (indirect) consequence of its deleterious effect on dorsal

hippocampus-dependent processing. Indeed, we previously showed that various (direct or indirect) *manipulations of the dorsal hippocampus affect in an opposite way tone and contextual fear conditioning* (Desmedt et al., EJN, 1999; Calandreau et al., J Neurosci, 2006; Calandreau et al., Learn & Mem, 2010; see also Calandreau et al., Learn & Mem, 2007), as well as amygdalar activity (Calandreau et al., 2006).

Otherwise, the deleterious impact of glucocorticoids/stress exposure on hippocampal function is attested by numerous reports on neuronal loss, decrease in dendritic branching, alterations in synaptic terminal structure, inhibition of neuronal regeneration, impairment of long-term potentiation (LTP) and in a reduction in the production of new neurons in the dorsal hippocampus (Abrous et al., Physiol Rev, 2005; Elzinga & Bremner, J Affect Disorder, 2002; Maggio & Segal, J Neurosci, 2007; Sapolsky, Science, 1996). This stress-induced impairment of hippocampal plasticity is likely to be responsible for the declarative memory deficit observed in PTSD.

To summarize, we suggest that CORT injection induces dysfunction of the dorsal hippocampus which thus promotes a *cognitive imbalance* in favor of an elemental (amygdala- and probably ventral hippocampus-based) tone fear conditioning.

A reference to this putative CORT-induced cognitive imbalance between tone and contextual fear memory has now been included in the Methods and Discussion sections of the revised version of the manuscript (cf. pp. 5-6 & 14-15).

4. Value of the tone ratio

“(Fig 1a, Tone Test) For the Laser OFF/Veh group, the % freezing during the Tone period is lower than % freezing during the Pre-T period. If so, why is the Tone Ratio shown as being greater than 0? This seems to be the case for all Tone Ratio figures shown in Figure 1.”

As written in the Method section of the original paper (pp. 18-19, see pp. 11-12 in the revised manuscript), the strength and specificity of the conditioned fear to the tone is “attested by a ratio that considers *the percentage of freezing increase to the tone with respect to a baseline freezing level (i.e., pre- and post-tone periods mean)*. Indeed, a strong and specific conditioned fear to the discrete tone CS implies a lower level of freezing when the shock is not expected (i.e. *before and after the tone presentation*) compared to the freezing level expressed during the tone presentation (high ratio value). The tone ratio is calculated as follows: $[\% \text{ freezing during tone presentation} - (\% \text{ pre-tone period freezing} + \% \text{ post-tone period freezing})/2] / [\% \text{ freezing during tone presentation} + (\% \text{ pre-tone period freezing} + \% \text{ post-tone period freezing})/2]$.”

As a consequence, even when the % freezing during the tone period is lower than % freezing during the pre-tone period (as in Laser OFF/Veh group), if the average of freezing percentage expressed during the *pre- and post-tone periods* is lower than the % of freezing during the tone period, the tone ratio is mechanically higher than zero.

5. “n sizes”

“Are the n sizes for the four experiments before or after histological verification? Proving both before and after histology n size information would be helpful.”

We thank the reviewer for this question which allows us to clarify this important aspect of the method. For all the experiments, the n sizes correspond to the number of animals *after histological verification*. Before histological control, all n sizes correspond to 10 mice.

This important information has now been included in the Method section of the revised version of the manuscript (cf. p. 10).

6. Quantification of freezing behavior.

“How was the freezing behavior quantified by a blind observer? Time sampling or duration?”

As indicated in the Method section of the original version of the manuscript (cf. p. 18, p. 11 in the revised manuscript), during the two memory tests, “animals were continuously recorded for off-line *second-by-second scoring of freezing* by an *observer blind of experimental groups*”.

7. β -cyclodextrin as vehicle instead of NaCl alone.

“Shouldn't the vehicle be β -cyclodextrin dissolved in .9% NaCl, rather than NaCl alone, to confirm that β -cyclodextrin is inert?”

In all our previous studies the complex of corticosterone with cyclodextrin, which allows dissolving this steroid in aqueous solutions, has been used in combination with fear conditioning (Kaouane et al., 2012; Revest, 2005; 2010; 2013). The fact that (i) most biochemical studies consider β -cyclodextrin as safe molecule used to form complexes with hydrophobic compounds and allow delivery of a variety of drugs, including hydrocortisone (GRAS Notice No. GRN 000155, GRAS Notice No. GRN 000074, GRAS Notice No. GRN 000046; Uekama et al., Chemical Reviews, 1998) and more importantly that (ii) systemic injections of CORT, in a range of doses that encompasses increase of CORT induced by several types of stress (from 0.75 to 10 mg/kg), induces PTSD-like memory in a dose-dependent manner (while concentration of β -cyclodextrin is almost constant) (Kaouane et al., Science, 2012), make very unlikely the idea that the observed effects of CORT injection on fear memory could be due to β -cyclodextrin.

Nevertheless, as suggested by the reviewer, we achieved an additional experiment in order to test this possibility.

As shown in Figure 2, CORT-injected mice display a PTSD-like memory profile whereas HBC-injected mice do not. Indeed, compared to HBC-injected mice, cort-injected mice display an abnormally high fear response to the tone (left, see also tone ratio, both $p < 0.05$) and lower levels of conditioned fear to the context (right).

All in all, these differences are almost similar to those observed between CORT-injected and NaCl-injected control mice and indicate that the effects of CORT cannot be due to its vehicle HBC. A reference to this observation has now been done in the legend of Figure 1 in the revised version of the manuscript (p. 18).

This figure may be included as supplementary figure in a final version of the manuscript upon request.

Figure 2. Injection of HBC cannot explain the deleterious effects of CORT on fear memory.

Reviewer 2:

1- Novelty of the findings

“(Remarks to the Author):

Alice Shaam Al Abed et al investigated roles of hippocampus in generation of PTSD-like memory and switching this memory to normal fear memory by trauma-contextualization and re-contextualization using optogenetics. The authors showed that optogenetic inactivation of hippocampus during the exposure to the cue is sufficient to generate PTSD-like memory while activation of hippocampus with CORT injection blocks the formation of PTSD-like memory. Interestingly, reactivation of traumatic memory by re-exposure to the context with the cue switched the PTSD-like memory to normal fear memory and this normalization was hippocampus-dependent. Furthermore, the authors showed the persistence of PTSD-like and normalized memories. Importantly, the findings support the importance of exposure therapy for PTSD that may normalize PTSD-like memory through reactivation of this memory. However, the novelty of findings in this manuscript seems to be not so much because the authors have already shown that the generation of PTSD-like memory is hippocampus-dependent using microinfusion of CORT into hippocampus (ref15).”

In our previous paper (Kaouane et al., Science, 2012), we indeed showed that intra-hippocampal corticosterone injection (like systemic corticosterone injection or restrain stress), performed after fear conditioning using a US of sufficiently high intensity, produces PTSD-like memory, that is both contextual amnesia and hypermnesia for a salient but irrelevant trauma-related cue (e.g. none predictive tone).

However, this was far to demonstrate that *contextual amnesia is the cause of, i.e. a necessary and sufficient condition for the development of pathological emotional hypermnesia* characteristic of PTSD. Although a critical role of contextual amnesia in PTSD-related emotional hypermnesia has been suggested by some clinicians in the field, this idea still remains controversial and was not tested until now. Here we demonstrate that contextual amnesia is both sufficient and necessary for the development and maintenance of PTSD-like emotional hypermnesia. **(i)** Inducing contextual amnesia (by *inhibition of the dCA1 of the hippocampus*) is sufficient to induce PTSD-like hypermnesia in conditions of moderate stress (without CORT injection). **(ii)** Contextual amnesia is necessary since, under traumatic stress (with CORT injection) **(a)** preventing contextual amnesia (by dCA1 activation) prevents the development of PTSD-like hypermnesia and **(b)** treating contextual amnesia (by re-exposure to the traumatic context) also treats PTSD-like hypermnesia.

Altogether, these new findings not only support the importance of exposure therapy for PTSD, as noted by the reviewer, but more specifically call for **(1)** promoting therapeutic approaches of PTSD centred on *trauma contextualization* and **(2)** developing pre-clinical studies centred on the *underlying hippocampal mechanisms*.

This novelty of the present findings has been emphasized by a slight modification of their synthesis in the conclusion of the revised version of the manuscript (cf. pp. 5-6).

2. How can hippocampal activation block generation of PTSD-like memory by CORT?

“Optogenetic activation of hippocampal CA1 in the presence of CORT blocked generation of PTSD-like memory. This observation is interesting. However, it still remains unclear why this artificial hippocampal activation blocks generation of PTSD-like memory by CORT. The authors should show or suggest how this activation could block the effect of CORT.”

We thank the reviewer for this important suggestion, which echoes the third issue raised by reviewer 1 (see point 3 in reply to reviewer 1, p. 3).

As indicated above, we previously showed that *manipulations of the hippocampus affect in an opposite way tone and contextual fear*, and that this seems to be mediated by an influence of the hippocampus on the amygdalar function (e.g. Calandreau et al., J Neurosci, 2006). Here, we can speculate that the impact of corticosterone on tone fear memory is an (indirect) consequence of its deleterious effect on dorsal hippocampus-dependent processing. We suggest that CORT injection

induces dysfunction of the dorsal hippocampus which thus promotes a *cognitive imbalance* in favor of an elemental (amygdala- and probably ventral hippocampus-based) tone fear conditioning.

At the cellular and molecular level, as written above (point 3, p. 3), numerous data have shown that stress, or corticosterone exposure, can impair the hippocampal plasticity (Abrous et al., *Physiol Rev*, 2005; Maggio & Segal, *J Neurosci*, 2007; Sapolsky, *Science*, 1996). We suggest that hippocampal activation might counteract some of the CORT-induced deleterious effects on the hippocampus and thus restore a cognitive imbalance in favor of the processing of contextual cues. Future studies will tell us which cellular and molecular mechanisms may be involved in this normalization of the hippocampal function. Whatever these mechanisms are, the present findings clearly show that the level of hippocampal activation is at the core of this key cognitive imbalance in the development of normal (contextual) vs. PTSD-like (cue-based) fear memory.

As requested by the Reviewer, our suggestion on how hippocampal activation could block the effect of CORT has now been included in the revised version of the manuscript (cf. pp. 5-6 & 14-15).

3. Optogenetic controls

“In figure1, several control experiments are required such as mice expressing GFP with laser and mice expressing ChR2/ArchT without laser.”

We thank the reviewer for this important request. The fact is that mice *expressing ChR2 or ArchT without laser* are already included in our study insofar as all our mice were submitted to virus injection (ChR2 or ArchT). Thus, our controls (Laser OFF) receiving systemic injection of NaCl or CORT had been previously submitted to hippocampal injection of ArchT and ChR2, respectively. This important methodological precision has now been included in the Method section of the revised manuscript (cf. table p. 16).

Regarding the control GFP, several observations indicate that light stimulation *per se* does not impact the observed freezing behavior. First, previous optogenetic experiments achieved in our laboratory clearly showed that the freezing behavior of control GFP mice was not impacted by light stimulation of the dCA1 neurons (cf. Sellami *et al.*, *PNAS*, 2017). Second, in the present study, the fact that optogenetic stimulation of dCA1 neurons produces *opposite effects on freezing* behavior (both to the tone and to the conditioning context) as a function of the virus injected (ChR2 vs. ArchT) makes it very unlikely, not to say impossible, any explanation of the behavioral effects observed by an action of the light *per se*.

The reference to our previous optogenetic experiments has now been included in the revised version of the manuscript (p. 14).

4. Memory updating processes potentially involved in the normalization of traumatic memory.

“Normalization of PTSD-like memory must be mediated by memory updating processes such as reconsolidation, extinction and/or re-learning. Although it is interesting to examine these mechanisms, at least, the authors should discuss about these possibilities.”

We also thank the reviewer for this suggestion. This discussion had been developed in a review published a few years ago (Desmedt et al., *Biol Psy*, 2015). In the present manuscript, for reasons of space limitations, we could not reproduce such discussion. However, as requested, the revised version of the manuscript now emphasizes such memory updating processes. One of the key processes that might be involved in the normalization of traumatic memory is the following:

Promising clinical approach relates to the “*reconsolidation*” of traumatic memory (Deacon & Abramowitz, *J Clin Psychology*, 2004). Numerous animal studies indicate that reactivated memory traces become transiently labile and thus sensitive to manipulations, as if they undergo a new transient phase of consolidation, dubbed “*reconsolidation*” (Dudai, *Annual Rev Psychol*, 2004; Nader *et al.*, *Nature Rev Neurosci*, 2000; Nader, *Nature*, 2003; Sara, *Nature Rev Neurosci*, 2000). Several cognitive and neurobiological factors have already been shown to promote either the strengthening of the original trace (i.e. its persistence) or its weakening in favor of a new “*memory trace*” (Dudai, *Annual*

Rev Psychol 2004; Dudai & Eisenberg, Neuron, 2004; Eisenberg *et al.*, Science, 2003). However, compared to *normal* fear memory, on which the reconsolidation process is generally studied, we may wonder whether a pathological (PTSD-related) fear memory can be sensitive to the same manipulation. Indeed, while frequent reactivation of a *normal* fear memory in safe contexts classically leads to a *reduction of this memory*, frequent intrusive reactivation of *PTSD-related memory* seems to contribute to its *persistence*. We suggest that reactivation of traumatic memory in PTSD would be incomplete. In contrast to normal fear memory, and as written in the original conclusion of our manuscript (cf. p. 6), “*the disconnection of PTSD-related memory from context-based representations would prevent re-evaluation of the traumatic event, thus strengthening the original memory trace*”^{6,19}. *In PTSD patients, intrusive reactivation of traumatic memories through flashbacks in safe contexts is frequent and indeed seems to contribute to the persistence of memory symptoms. In contrast, the re-appraisal of the traumatic event in the light of its contextual representation would promote its integration into the declarative memory system, leading thereby to recovery from PTSD-related memory*^{6,19}”.

“The therapeutic goal is not to erase the declarative memory of the traumatic event itself. This would worsen the situation not only by making the patients amnesic with regard to such a critical event of his/her past life experience but also by strengthening the implicit and automatic traumatic hypermnesia that precisely appears as a consequence of declarative amnesia in PTSD (Yovell *et al.*, CNS Spectrums, 2003). In contrast, the goal is rather to eliminate the traumatic/pathological component of the fear memory by promoting the hippocampal processing, and thus the declarative memory of the traumatic event” (cf. Desmedt *et al.*, Biol Psy, 2015).

The possibility that *reconsolidation of traumatic memory (based on its re-appraisal in the light of its contextual representation)* may mediate its normalization has been emphasized in the conclusion of the revised manuscript (cf. p. 6).

In summary, we have fully addressed the concerns of the reviewers and incorporated the suggested changes in the revised manuscript as much as possible.

Reviewers' Comments:

Reviewer #1:

Remarks to the Author:

While the revised manuscript has been improved, my comments 1, 2, and 4 (as originally stated below) have not been properly addressed.

1. The pre-tone freezing levels during the tone test in a familiar chamber are comparably high in all groups (e.g., Fig. 1a,b; Fig. 3), which would indicate generalized fear between trained and familiar chambers (both are round with 20 cm diameter). If so, shouldn't there be some re-contextualization going on during the tone test?

The re-contextualization explanation does not work because the pre-T freezing is comparably high between familiar vs. trained chambers. To rigorously test the authors' claim, I strongly recommend that a different familiar chamber (i.e., a novel chamber where there is minimal context generalization) be used.

2. The post-training cort injection producing PTSD-like effects require some explanation. For example, is the cort effect due to its action on dCA1 cells since the dCA1 inhibition during fear conditioning mimics post-training cort effect, whereas the dCA1 activation offsets post-training cort effect? If so, this should be tested by applying optogenetic inhibition, not only 5 minutes before and during fear conditioning (as shown in Fig 1b,c) but also right after fear condition.

If post-training optogenetic inhibition manipulation does not mimic post-training cort effects on producing PTSD-like effects, this would argue against the authors claim. I cannot follow the authors' logic getting around this issue.

4. (Fig 1a, Tone Test) For the Laser OFF/Veh group, the % freezing during the Tone period is lower than % freezing during the Pre-T period. If so, why is the Tone Ratio shown as being greater than 0? This seems to be the case for all Tone Ratio figures shown in Figure 1.

The authors state that the tone ratio is calculated as: $[\% \text{ freezing during tone presentation} - (\% \text{ pre-tone period freezing} + \% \text{ post-tone period freezing})/2] / [\% \text{ freezing during tone presentation} + (\% \text{ pre-tone period freezing} + \% \text{ post-tone period freezing})/2]$.

What is the rationale (or basis) or dividing by 2? Please provide literature supporting the tone ratio calculation.

5. Are the n sizes for the four experiments before or after histological verification? Providing both before and after histology n size information would be helpful.

The revised manuscript (below) does not show the n sizes before excluding the animals based on histological analysis. :

The group sizes (after histological verification) were as follows:

- Experiment 1 [optogenetic inhibition/activation of dCA1 (vs. dCA2 and dCA3) during fear conditioning (vs. before conditioning)]: n = 8 to 17 mice per group.
- Experiment 2 (trauma "re-contextualization"): n = 9 to 10 mice per group.
- Experiment 3 (optogenetic inhibition of dCA1 during trauma "re-contextualization"): n = 7 mice per group.
- Experiment 4 (assessment of the persistence of PTSD-like and "re-contextualized" fear memory on day 30): these mice are the same as in experiments 2 and 3.

Reviewer #2:

Remarks to the Author:

The authors adequately responded to concerns.

Description of revisions and reply to the reviewer' concerns:

General:

We were pleased to learn the present findings were considered of interest for a potential publication in Nature Communication after an additional revision of the manuscript. While Reviewer 2 considered that we adequately responded to his/her concerns, Reviewer 1 still raised 4 concerns that need to be addressed.

The requested changes have been included in the manuscript (cf. text highlighted in yellow in the revised manuscript) as requested by the Reviewer's recommendations.

Our point-by-point responses to these last concerns of Reviewer 1 are detailed below:

Reviewer 1:

1. The re-contextualization of traumatic memory (in the traumatic context) must imply discrimination between the traumatic and familiar context.

“The pre-tone freezing levels during the tone test in a familiar chamber are comparably high in all groups (e.g., Fig. 1a,b; Fig. 3), which would indicate generalized fear between trained and familiar chambers (both are round with 20 cm diameter). If so, shouldn't there be some re-contextualization going on during the tone test?”

The re-contextualization explanation does not work because the pre-T freezing is comparably high between familiar vs. trained chambers. To rigorously test the authors' claim, I strongly recommend that a different familiar chamber (i.e., a novel chamber where there is minimal context generalization) be used.”

Here we believe that there is no fundamental disagreement with the Reviewer but essentially misunderstanding from both sides, likely due to our imprecisions/ language errors.

The reviewer suggests that the observed normalization of traumatic memory on day 4 after re-exposure to the tone in the conditioning context can unlikely be explained by a re-contextualization of traumatic memory in the conditioning context. Because similar levels of freezing are “being observed” in the conditioning and familiar contexts, mice would not discriminate these two contexts. In other words, the normalization of traumatic memory being specifically observed after re-exposure (to the tone) in the *conditioning* context, but not in the familiar context, any re-contextualization explanation would imply a necessary discrimination between the two contexts.

We thank the reviewer for the new formulation of this important concern. On the basis of this criticism, we have first corrected the description of the two contexts used in our study, which could initially suggest that these two contexts were similar when they are in fact very different. Second, we have deepened the analysis of figures 1 and 2 and direct comparison of figures **1a left** and **right**, but even more of **2a** and **2b**, establishes that all groups, including the CORT-injected one, do discriminate the two contexts, thus invalidating the Reviewer's concern.

- First, these two contexts are very different from each other as indicated for instance by the nature of the *floor* (opaque PVC floor vs. footshock grid) or the *smell* present in the chamber (one is cleaned with 1% acetic acid, the other with 70% ethanol) (see Methods p. 11). The *shapes* of these two chambers were not properly described in the previous version of the manuscript, which led to erroneously consider that both chambers could be round. In fact, although similar in size, the familiar chamber is *round* (20 cm diameter), whereas the conditioning chamber is *squared* (20 cm x 20 cm). We have thus specified and corrected the description of these two chambers accordingly.

- Second, as indicated in our previous reply, the relatively high pre-tone freezing levels (\approx 24, 22 and 30% in OFF/CORT, dCA1 inhib/Veh and OFF/Veh mice, respectively), classically observed in all groups in the familiar context (**Figure 1a, left**), are likely due to the specific (anxiogenic) conditions

of this memory test which is the first test occurring after the aversive conditioning (cf. our previous reply). Furthermore, these relatively high freezing levels are *nevertheless lower than those expressed in the conditioning chamber* (**Figure 1a, right**: $\approx 38, 50$ and 70% in OFF/CORT, dCA1 inhib/Veh and OFF/Veh mice, respectively). The fact that freezing levels are significantly different between familiar and conditioning context in all groups (OFF/CORT: $F_{1,12} = 6.295$; $p=0.0275$; dCA1 inhib/Veh: $F_{1,12} = 32.748$; $p<0.0001$; OFF/Veh mice: $F_{1,14} = 100.188$; $p<0.0001$) is indicative of a differential processing of the context depending on its nature (aversive vs. familiar/safe).

- Third, this difference between the levels of freezing expressed in the familiar and conditioning contexts is even more pronounced on day 3 when animals are re-exposed to the tone in the conditioning context (see the first 2 min prior to tone presentation) or in the familiar context (**Figure 2a vs. 2b left**). Indeed, CORT-injected animals display $\approx 38\%$ of freezing in the conditioning context but only 20% in the familiar context ($F_{1,17} = 7.166$; $p=0.0159$). This key observation indicates that, once the first memory test underwent (test whose conditions likely explain a relatively high basal level of freezing on day 2), CORT-injected animals do discriminate perfectly well between the two contexts on day 3, and *while their level of freezing expressed in the conditioning context is dramatically impaired* compared to the control condition ($\approx 38\%$ vs. 70% in Veh-injected animals) *it is nevertheless significantly higher than their level of freezing expressed in the familiar context* ($\approx 20\%$). This key observation excludes the hypothesis of a complete generalization of contextual fear and explains the normalization of PTSD-like memory by trauma re-contextualization during the tone-induced evocation of the traumatic memory *in the traumatic context specifically*.

Hence, not only the fact that the re-exposure to the tone *in the conditioning context* is the only condition inducing a normalization of traumatic memory on day 4 (see Figure 2), but also that animals do discriminate the two contexts (traumatic vs. familiar ones) before “re-contextualisation”, allow us to definitively conclude that the normalization of traumatic memory results from its re-contextualization. The observations relative to this contextual discrimination have been synthesized in the last revised version of the manuscript (cf. Discussion section, p. 5) and the related statistical analyses have been included in the legends of Figures 1 & 2 (pp. 19 & 20).

2. The post-training optogenetic inhibition manipulation does not mimic post-training cort effects on fear memory.

“The post-training cort injection producing PTSD-like effects require some explanation. For example, is the cort effect due to its action on dCA1 cells since the dCA1 inhibition during fear conditioning mimics post-training cort effect, whereas the dCA1 activation offsets post-training cort effect? If so, this should be tested by applying optogenetic inhibition, not only 5 minutes before and during fear conditioning (as shown in Fig 1b,c) but also right after fear conditioning.

If post-training optogenetic inhibition manipulation does not mimic post-training cort effects on producing PTSD-like effects, this would argue against the authors claim. I cannot follow the authors’ logic getting around this issue.”

Here, we disagree with the Reviewer’s point that post-conditioning optogenetic inhibition of the dCA1 cells for 5 minutes should absolutely reproduce the PTSD-like effect of post-conditioning peripheral Cort injection in order to be able to conclude that Cort induces PTSD by having a deleterious impact on hippocampal function. We try to explain why we disagree in the followings:

As written in our previous letter, we do believe that the effect of post-training peripheral injection of CORT is due to its deleterious impact on the dorsal hippocampus-dependent formation of declarative/contextualized fear memory. The main support for this view is the fact that CORT injected directly into the dorsal hippocampus has the same (PTSD-like memory inducing) effect as peripheral injection of CORT as shown in our original paper (Kaouane et al., Science, 2012). In fact, when combined with fear conditioning using a US of sufficiently high intensity (as in the present study), post-training corticosterone injection mimics traumatic stress. This is further confirmed by the fact that the deleterious effect of CORT (intra-dHPC or systemic injections) can be more physiologically reproduced by a post-training restraint stress (Kaouane et al, Science, 2012). In contrast, when combined with fear conditioning using a US of low intensity, post-training CORT mimics moderate

stress and in that case, promotes/enhances dHPC-dependent contextualized fear memory (as we previously showed in Revest et al, Nat. Neurosci., 2005, and Kaouane et al, Science, 2012).

Now, the post-training optogenetic inhibition of dCA1 does not necessarily recapitulate the deleterious effect of post-training injection of CORT on the dHPC-dependent contextualized fear memory. Drastic differences between the two manipulations may explain their different outcomes.

- First, these two manipulations have extremely *different timings*. Post-training administration of CORT might affect the *consolidation* of dHPC-dependent memory: in other words, this effect can extend over several hours after conditioning. In contrast, the optogenetic inhibition of dCA1 cells lasts 4 minutes in our study. Since (i) the *precise time window* during which the optogenetic manipulation should be performed during the (24 hours) retention interval is difficult to predict and (ii) optogenetic manipulation *cannot be performed during the entire 24h retention interval*, it seems extremely difficult, not to say impossible, to mimic the CORT-induced memory manipulation by the optogenetic manipulation of dCA1 cells.

- Second, stress or CORT administration has numerous well-known deleterious impacts on the hippocampus that may affect memory consolidation, e.g. neuronal loss, decrease in dendritic branching, alterations in synaptic terminal structure, inhibition of neuronal regeneration, impairment of long-term potentiation (LTP), and reduction in the production of new neurons in the hippocampus (Abrous et al., 2005; Elzinga & Bremner, 2002; Sapolsky, 1996). Again, a time limited post-training optogenetic inhibition of dCA1 cells does not recapitulate these numerous and more or less long-lasting CORT-induced functional and structural hippocampal alterations.

- Third, CORT-induced manipulation of memory consolidation may involve alterations extending beyond the dCA1 sub-region of the hippocampus. In our original paper, CORT injection targeted the dorsal hippocampus. Thus, even if an alteration of the dCA1 is supposed to be critically involved in the CORT-induced contextual memory alteration, one cannot exclude the additional involvement of other altered HPC sub-regions, as for instance the DG.

For all these reasons, we initially chose not to manipulate the dCA1 (also *after* fear conditioning. The additional optogenetic experiment in which animals were submitted to dCA1 inhibition *right after* fear conditioning confirmed that such time-limited inhibition of dCA1 cells could not recapitulate the deleterious effects of post-training CORT injection on memory consolidation (see previous reply). In contrast, the fact that, (1) in humans, a deficit in mental narrative of the event *during* the trauma seems to be a risk factor for the development of PTSD (Brewin, 2011), and that (2) this time window could be entirely covered by our optogenetic approach incited us to choose the optogenetic manipulation of the dCA1 *during* fear conditioning. Results clearly show that while optogenetic inhibition of dCA1 *right after* fear conditioning cannot mimic CORT-induced formation of PTSD-like memory, the same inhibition *during* trauma can produce such traumatic memory and that contextual amnesia is at the core of this maladaptive fear memory. Future studies will tell us which post-traumatic CORT-induced hippocampal alterations are key mechanisms for the development of PTSD-like memory.

This critical point had been synthesized in the Method section of the revised manuscript and has been re-emphasized in the last revised version of the manuscript (cf. p. 15-16).

4. Calculation of tone ratio

“(Fig 1a, Tone Test) For the Laser OFF/Veh group, the % freezing during the Tone period is lower than % freezing during the Pre-T period. If so, why is the Tone Ratio shown as being greater than 0? This seems to be the case for all Tone Ratio figures shown in Figure 1.

The authors state that the tone ratio is calculated as: $[\% \text{ freezing during tone presentation} - (\% \text{ pre-tone period freezing} + \% \text{ post-tone period freezing})/2] / [\% \text{ freezing during tone presentation} + (\% \text{ pre-tone period freezing} + \% \text{ post-tone period freezing})/2]$.

What is the rationale (or basis) or dividing by 2? Please provide literature supporting the tone ratio calculation.”

As written in our previous reply as well as in the Method section of the original paper (pp. 18-19, see pp. 12 in the revised manuscript), the strength and specificity of the conditioned fear to the tone is

“attested by a ratio that considers *the percentage of freezing increase to the tone with respect to a baseline freezing level (i.e., pre- and post-tone periods average)*. Indeed, a strong and specific conditioned fear to the discrete tone CS implies a lower level of freezing when the shock is not expected (i.e. *before and after the tone presentation*) compared to the freezing level expressed during the tone presentation (high ratio value).

Now, in the formula, what is necessarily divided by 2 is the sum of freezing percentage *before* (block of 2 min: pre-tone period) and freezing percentage *after* (block of 2 min: post-tone period) tone presentation in order to get an *average percentage of freezing during a 2-min baseline period* that will be thus compared to the percentage of freezing observed during the 2 min-block presentation of the tone.

As written previously, if the average of freezing percentage expressed during the *pre- and post-tone periods* is *lower* than the % of freezing expressed during the tone period, the tone ratio will be mechanically higher than zero.

This ratio is routinely used in our laboratory since we noticed that the absolute level of freezing expressed *during* the tone presentation only was a bad behavioral indicator of tone fear conditioning *per se*. Indeed, animals can display very high level of freezing to the tone but similar high levels of freezing before and after tone presentation, which is thus indicative of generalized fear responses but does not at all prove any specific conditioned fear to the tone. Reciprocally, in some conditions, animals can express relatively low levels of freezing to the tone but significantly lower levels of freezing before and after tone presentation, which is thus indicative of a significant conditioned fear to the tone.

This ratio has initially been published in the following paper, which is cited in both the main text of the manuscript (Discussion section, p.6) and in the Method section (p.11) (see Reference 21, p.8):

Calandreau L., Trifilieff P., Mons N., Costes L., Marien M., Marighetto A., Micheau J., Jaffard R. & Desmedt A. (2006) Extracellular hippocampal acetylcholine level controls amygdala function and promotes adaptive conditioned emotional response. *Journal of Neuroscience*, 26:13556-13566.

5. N sizes before and after histological verification.

“Are the n sizes for the four experiments before or after histological verification? Providing both before and after histology n size information would be helpful.

The revised manuscript (below) does not show the n sizes before excluding the animals based on histological analysis. :

The group sizes (after histological verification) were as follows:

- *Experiment 1 [optogenetic inhibition/activation of dCA1 (vs. dCA2 and dCA3) during fear conditioning (vs. before conditioning)]; n = 8 to 17 mice per group.*
- *Experiment 2 (trauma “re-contextualization”): n = 9 to 10 mice per group.*
- *Experiment 3 (optogenetic inhibition of dCA1 during trauma “re-contextualization”): n = 7 mice per group.*
- *Experiment 4 (assessment of the persistence of PTSD-like and “re-contextualized” fear memory on day 30): these mice are the same as in experiments 2 and 3.”*

In order to clarify this methodological issue, a table recapitulating all N sizes *before* (starting number) and *after* (final number) histological verification for each experimental condition has now been included in the Method section of the revised manuscript (cf. p. 10).

In summary, we have fully addressed the concerns of the reviewer and incorporated the suggested changes in the revised manuscript as much as possible.

Reviewers' Comments:

Reviewer #1:

Remarks to the Author:

1. The authors have adequately described the differences between familiar (a 20 cm diameter round) chamber vs. conditioning (a 20 x 20 cm square) chamber. However, now all figures show familiar and conditioning chambers as being square (if I am not mistaken, I think they were all shown as round previously). I recommend familiar chambers to be represented as round and conditioning chambers depicted as square in all figures.

Related to the above, the authors state on Pg. 11: "A squared conditioning chamber (20 x 20 cm)...The floor of the chamber consisted of 30 stainless-steel rods (5 mm diameter), spaced 5 mm apart..." I do not see how 30 rods (5 mm diameter, 5 mm apart) can fit inside 20 x 20 cm square chamber.

2. I remain steadfast for the third time that the immediate post-training cort injection producing PTSD-like effects require some explanation. The claim that post-training cort is causally involved in PTSD-like memory formation via inducing contextual amnesia needs to be supported by facts and not mere conjectures. Whether the authors' argument that "the post-training optogenetic inhibition of dCA1 does not necessarily recapitulate the deleterious effect of post-training injection of cort on the dHPC-dependent contextualized fear memory" is valid or not won't be known unless such experiment has been performed. If cort effects are due to affecting the memory consolidation process that occurs during a critical time window after aversive experience, this can be tested optogenetically as was done during fear conditioning. I believe such post-training manipulations are typical in memory consolidation experiments.

Description of revisions and reply to the reviewer' concerns:

General:

We were pleased to learn the present findings were considered of interest for a potential publication in Nature Communication after a final revision of the manuscript. Reviewer 1 still raised 2 concerns, one of which can be resolved by a suggestion from the Editor, while the second one is a point of methodological clarification. The requested changes have been included in the manuscript (cf. text highlighted in yellow in the revised manuscript) as requested by the Reviewer's recommendations.

Our point-by-point responses to these last concerns are detailed below:

Reply to the Editor and Reviewer 1:

1. Relationship between the deleterious effect of post-training injection of corticosterone on the dHPC-dependent contextualized fear memory and the effect of dCA1 inhibition during conditioning.

Reviewer 1: "I remain steadfast for the third time that the immediate post-training cort injection producing PTSD-like effects require some explanation. The claim that post-training cort is causally involved in PTSD-like memory formation via inducing contextual amnesia needs to be supported by facts and not mere conjectures. Whether the authors' argument that "the post-training optogenetic inhibition of dCA1 does not necessarily recapitulate the deleterious effect of post-training injection of cort on the dHPC-dependent contextualized fear memory" is valid or not won't be known unless such experiment has been performed. If cort effects are due to affecting the memory consolidation process that occurs during a critical time window after aversive experience, this can be tested optogenetically as was done during fear conditioning. I believe such post-training manipulations are typical in memory consolidation experiments."

We maintain that it is extremely difficult, not to say impossible, to perform an inhibition of dCA1 cells *during the entire time window* of the post-training memory consolidation with optogenetics. Then, in order to address the issue raised by the Reviewer, it is *"important to show that the post conditioning cort injection effects on PTSD memory are linked to those resulting from the optogenetic manipulation of hippocampal activity during conditioning."*, as the Editor said. Thus, he suggested that *"the authors should provide some **molecular or physiological signature that shows a significant relationship between the two manipulations.**"* We followed this recommendation as detailed in the followings.

In our first letter in reply to Reviewer 1 (August 2019) we showed that the post-training optogenetic inhibition of dCA1, performed upon the request of the Reviewer, could not reproduce the deleterious effect of post-training injection of CORT on fear memory. In our second letter (October 2019), we gave the numerous possible reasons why such post-training optogenetic manipulation could not reproduce the deleterious effect of post-training CORT injection, among them the most obvious one being the extremely *different timings* of the two manipulations (i.e. a few minutes vs. a few hours).

Now, in contrast to the optogenetic manipulation *right after* fear conditioning, the optogenetic inhibition *during* conditioning (which implies a *time window that could be entirely covered by our optogenetic approach*) can mimic the effect of post-training CORT injection on fear memory. Namely, both manipulations induce PTSD-like memory alterations: a contextual amnesia and an emotional hypermnnesia under the form of abnormal fear to a salient but irrelevant cue. Our interpretation is that both manipulations produce, although in a different manner/timing, some alteration of hippocampal function that is crucial to form a contextual memory of a stressful event. Consequently, both

manipulations induce a contextual amnesia together with an emotional hypermnesia resulting from the contextual amnesia. As suggested by the Editor, in order to support our interpretation, we now provide molecular evidence (c-Fos) that both manipulations reduce hippocampal activity. More precisely, while it is known that the optogenetic inhibition of dCA1 neurons (based on AAV-CaMKII α -ArchT-GFP in C57BL/6J) in a fear conditioning paradigm significantly reduces c-Fos expression in this hippocampal sub-region (Wilmot et al., 2019: *cf. Figure 1 below*), we previously showed that the post-training injection of CORT also results in a significant reduction of c-Fos expression in the dHPC, and in particular in the dCA1 (Kaouane et al., Science, 2012: *cf. Figure 2 below*). Hence, as suggested by the Editor, these observations do provide evidence that the two manipulations used, both leading to PTSD-like memory, share a *similar molecular signature* of contextual amnesia which is at the core of this maladaptive fear memory.

Figure 1: reduction of c-Fos expression in the dCA1 after optogenetic inhibition of dCA1 cells in a fear conditioning paradigm. * $p < 0.05$ relative to control (from Wilmot et al., Front Behav Neurosci, 2019, Fig.1).

Figure 2: reduction of c-Fos expression in the dCA1 in CORT-injected mice that developed a PTSD-like memory compared to aCSF-injected mice submitted to contextual or tone fear conditioning. # $p < 0.05$: aCSF vs. CORT (from Kaouane et al., Science, 2012, Fig.3).

We hope that this molecular signature of contextual amnesia can now convince the reviewer that both post-training CORT injection and the optogenetic inhibition of dCA1 during conditioning have *similar deleterious effect on the hippocampus-dependent contextual memory* and produce thereby the PTSD-like emotional (cue-based) hypermnesia.

This important observation, together with the new key associated reference, has been included in the Method section of the last revised version of the manuscript (cf. p. 15 & p.18).

2. Description and illustration of conditioning vs. familiar chambers.

Reviewer 1: "The authors have adequately described the differences between familiar (a 20 cm diameter round) chamber vs. conditioning (a 20 x 20 cm square) chamber. However, now all figures show familiar and conditioning chambers as being square (if I am not mistaken, I think they were all shown as round previously). I recommend familiar chambers to be represented as round and conditioning chambers depicted as square in all figures.

Related to the above, the authors state on Pg. 11: "A squared conditioning chamber (20 x 20 cm)...The floor of the chamber consisted of 30 stainless-steel rods (5 mm diameter), spaced 5 mm apart..." I do not see how 30 rods (5 mm diameter, 5 mm apart) can fit inside 20 x 20 cm square chamber."

-First, as requested by the reviewer, in order to facilitate the distinction between the two chambers used in the present study, the *familiar* chamber has been represented as *round* and the *conditioning* chamber depicted as *square* in all figures.

-Second, we thank the Reviewer for his remark relative to the conditioning chamber surface. Indeed, the previous description of the conditioning chamber was unfortunately linked to a previous study. Now, as requested, the area of the squared conditioning chamber actually used in the present study (24 x 24 cm) as well as the *number* (19) and *width of the rods* (3 mm diameter, spaced of 1 cm apart) of the footshock grid have been corrected and are properly described in the last revised version of the manuscript (cf. Material & Methods, pp. 11).

In summary, we have fully addressed the last concerns of the reviewer and incorporated the suggested changes in the revised manuscript. We thank the Editor and the Reviewer for helping us to improve the paper